# Dynamic Transmissions and Volatility Spillovers between Global Price and U.S. Producer Price in Agricultural Markets

**Jin Guo** *,[†] 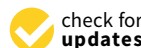 **and Tetsuji Tanaka** [†]

Department of Economics, Setsunan University 17-8 Ikedanaka, Neyagawa 572-8508, Japan;
tetsuji.tanaka@econ.setsunan.ac.jp
* Correspondence: kaku@econ.setsunan.ac.jp
† These authors contributed equally to this work.

**Abstract:** A considerable number of studies have examined the relationship between global prices and local prices in food-importing nations, but the linkages between international prices and the producer prices of large agricultural exporters have been largely ignored. This paper analyzes the connections between world prices and U.S. producer prices in the wheat, soybeans, and corn markets using a vector error correction generalized autoregressive conditional heteroscedastic model with a multivariate Baba-Engle-Kraft Kroner specification (VECM-GARCH-BEKK) and cross-correlation function (CCF). Our findings indicate firstly that a long-run equilibrium relationship exists between international and U.S. producer prices for the three agricultural crops. It also finds a significant bidirectional causality-in-mean and causality-in-variance between international and U.S. producer prices for these crops. Finally, the empirical results suggest that international wheat and corn prices play a leading role in U.S. local markets in return transmissions and that U.S. wheat price can be considered to be a leading indicator of the global wheat price in volatility transmissions.

**Keywords:** producer price; price transmission; VECM-GARCH-BEKK; Granger causality; volatility spillovers

## 1. Introduction

A vast body of literature has sought to delineate the price relationships between miscellaneous markets. For instance, Kouyaté and von Cramon-Taubadel (2016) showed that approximately 500 recently published papers could be found with the search phrase "price transmission" on AgEcon-Search. Many of these extant research projects analyze market connectivity within developing nations (e.g., Van Campenhout 2007; Myers 2008; Lutz et al. 2006; Rashid 2004; Abdulai 2000; Baulch 1997), while relatively fewer articles scrutinize price co-movements from global to local markets (e.g., Minot 2011; Mundlak and Larson 1992; Quiroz and Soto 1995; Conforti 2004; Robles and Torero 2010; Guo and Tanaka 2019), and most of these studies employed error correction models (ECMs). Ceballos et al. (2017) produced the first work on international price volatility passthroughs in the cereal industry and used a bivariate T-GARCH-BEKK model to investigate price volatility transmissions from the world market to grain-importing regional markets. Thus, past studies on this issue have highlighted food-importing nations rather than food suppliers, despite the fact that food suppliers are a vital component in pricing as well. For example, when U.S. soybean exports to China fell by 75% from September 2018 to May 2019 compared with the same nine-month period in the previous year due to the escalation of the U.S.-China trade war, some farming operations in South Dakota were destroyed by the lower producer price transmitted via the international price (Plume 2019; Newburger 2019). Such regulations in international or domestic markets can worsen market efficiency, generating extra margins between international, import or export, wholesale, and retail prices.

Copious literature inspects the market value chain and the vertical price transmission of agricultural commodities, namely the relationships between farmgate, wholesale, retail, and international prices. Any exporting firm needs to bear a fixed cost for building a distributional network overseas, and wholesalers connect farming operators to importers (Akerman 2010). Thus, the value chain structure takes several steps to deliver products to consumers, and the price connectivity of each stage has been studied by many articles with ECMs, many of which concentrate on the symmetry of price behavior (e.g., Rezitis and Tsionas 2019; Zingbagba et al. 2019; McLaren 2015; Usman and Haile 2017; Abdulai 2002; Worako et al. 2008).

Nonetheless, these studies do not particularize the impacts of the producer price of a large exporter on the international price or vice versa. Although Ghoshray (2008) analyzes the interconnectivity of rice export prices between two great exporting countries of rice, Thailand, and Vietnam, they do not examine the relationships between international and domestic markets. Wheat, corn, and soybean are vitally important commodities, but there is no information available on whether the producer prices of the U.S., one of the largest exporters, lead international prices or whether the causal relationships involving these crops are uni- or bi-directional with regards to the markets.

There is a growing demand for investigations of producer prices and relevant factors resulting from climate change, which exacerbates the fluctuations in agricultural crop production and jeopardizes global food security. Climate change in the form of extreme weather patterns has resulted in greater yield variability in recent years (Ünal et al. 2018). For example, the poor harvests in Australia and Ukraine in 2007 may have worsened the 2008 food crisis, and Russia's poor harvest in 2010 unsettled the global wheat market (Tanaka et al. 2012; Welton 2011). To make matters worse, future climate change will be aggravated by the migration of crop production to other areas (Reimer and Li 2009). However, past studies have utterly underestimated the importance of analyzing the interconnection between producer prices of countries producing large amounts of exports and global prices in the agricultural sector, particularly in the light of the fact that an ample number of published articles have found price spillover effects from international to local food-importing markets. Moreover, it has not been fully elucidated as to whether or not international price is actually transmitted to producer price in countries exporting large amounts of agricultural products despite the fact that the opposite directional causality, namely from domestic prices in exporting countries to global prices. For instance, it is understood intuitively from historical events such as the poor Russian wheat harvest in 2010.

This article contributes to establishing the relationship between producer prices in the U.S., one of the greatest agricultural exporters, and international crop prices by employing rigorous econometric methods. In particular, we identify the magnitude and speed of price transmissions, both from global to regional markets and from local to world markets. There is a remarkably limited number of articles examining international–producer price linkages, despite the vast literature on price transmissions from global to food-importing markets. The findings from our analysis can be utilized to calm tempestuous global food markets if U.S. farmers could obtain early and precise climate or weather forecasting information to stabilize their crop production. In addition, to this point, it is not fully understood whether exporters' producer prices are uni-directionally or bi-directionally connected to international prices.

The outline of the paper is as follows. Section 2 provides detailed information on the data used for our experiments. Then the methodology applied is introduced in Section 3. Section 4 explains the estimation results and Section 5 presents the results of the robustness tests. Finally, Section 6 concludes the article.

## 2. Data and Sample Statistics

We used monthly producer price indices for wheat, soybean, and corn from the U.S. Bureau of Labor Statistics and quoted monthly international prices for these crops from the International Monetary Fund (IMF) Commodity Prices spanning from January 1980 to December 2017, as the IMF

Commodity Prices are broadly utilized as global commodity prices.[1] The producer prices of wheat, corn, and soybean are indices standardized to 1982 and have been seasonally adjusted. In particular, No. 1 hard red winter wheat from Kansas City, No. 2 yellow corn from the Gulf of Mexico, and No. 2 yellow soybean future contracts from Chicago are used as the international prices of wheat, corn, and soybean, respectively.

Logarithmic transformations of each price series are used in the empirical analysis. More specifically, continuously compounded price returns are computed as $\ln(P_t / P_{t-1}) = \ln P_t - \ln P_{t-1}$, where $\ln P_t$ is the monthly logarithmic price for wheat, soybean, or corn for the international and U.S. producer markets. The developments in continuously compounded returns for each price series are plotted in Figure 1. Examining Figure 1, we make some interesting observations. First, all the price returns exhibit time-varying volatility, i.e., the volatilities are large in one period but small in another period. Second, the volatility tends to present a clustering behavior, i.e., periods of high (low) volatility tend to be followed by periods of high (low) volatility. Third, it is also interesting that some price returns (e.g., wheat and soybean) fluctuated dramatically during the 2007–2008 food crisis.

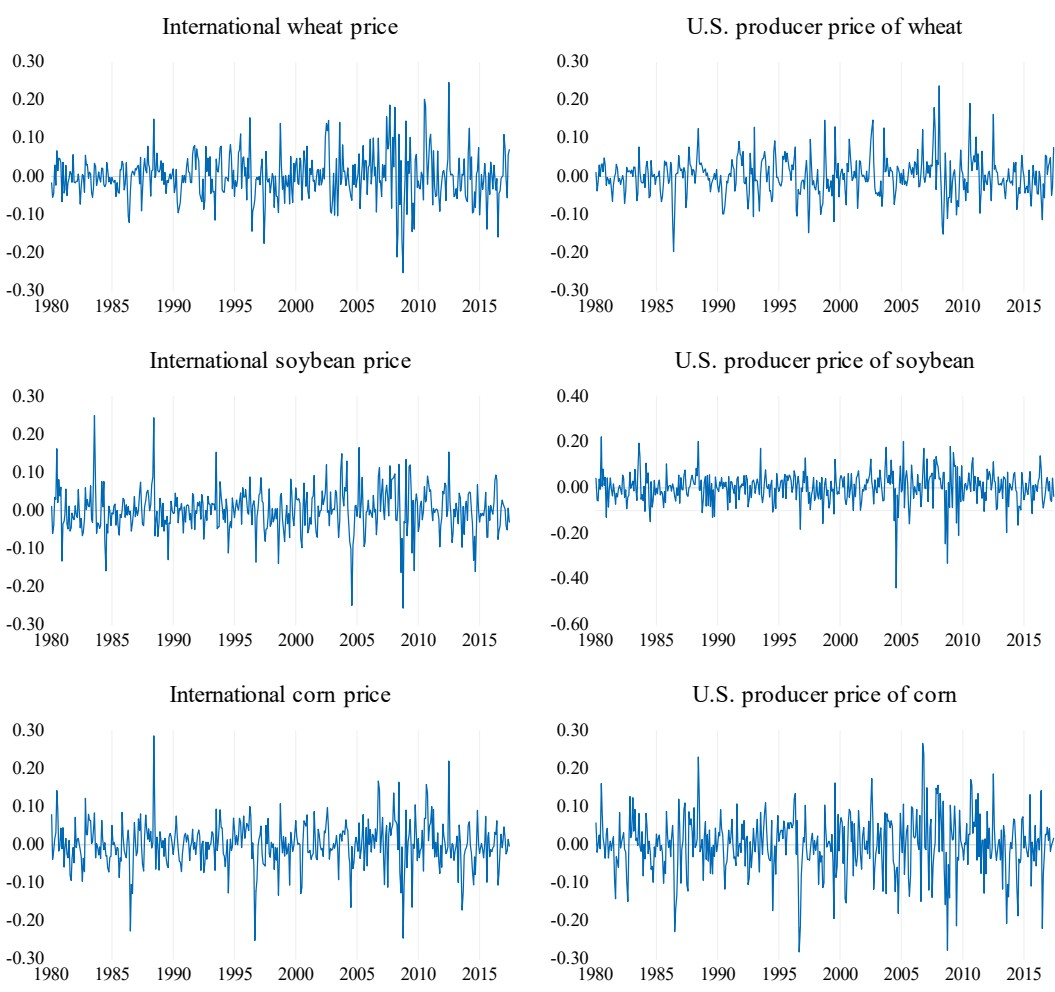

**Figure 1.** Development in returns for wheat, soybean, and corn prices.

Table 1 reports the descriptive statistics for the returns on international and U.S. producer prices for wheat, soybean, and corn. The data in this table reveal that most of the mean and median returns are positive during the study period, suggesting a rise in these prices either in the international market

---

or the U.S. local market. Moreover, it is evident that the standard deviation of the U.S. producer price for corn is relatively higher than those for the other commodities, suggesting that extreme changes tend to occur more frequently for this price return. Further, prior to empirical analysis, it is important to examine the stationarity of each price series to ensure their appropriateness for model estimation. The augmented Dickey-Fuller (ADF)[2] and Kwiatkowski-Phillips-Schmidt-Shin (KPSS)[3] unit root tests are applied to all the price returns. The results of the unit root tests presented in Table 1 indicate that the ADF test rejects the null hypothesis of nonstationary, while the KPSS test cannot reject the null hypothesis of a stationary series for all commodities. These results suggest that all variables are stationary in their first log-differenced forms[4] and integrated order 1 (*I* [Akerman]). Additionally, Van Dijk et al. (2005) suggest pre-testing for structural breaks before examining causality. Following this, Bai and Perron (2003) structural change test, which allows for the simultaneous estimation of unknown multiple structural breaks, is applied to identify the structural breakpoints in each wheat price series.[5] The results of the structural breaks test in Table 1 show that there are no structural breaks in all the price returns.

[1]    The Federal Reserve Economic Data also quotes the dataset as world prices.

**Table 1.** Summary statistics and unit root tests for price returns.

|  | Mean | Median | Std. Dev. | ADF Test | KPSS Test | Structural Break Tests |
|---|---|---|---|---|---|---|
| International wheat price | 0.000 | −0.002 | 0.060 | −13.718 *** (1) | 0.087 (3) | No break |
| U.S. producer price of wheat | 0.000 | 0.000 | 0.050 | −11.296 *** (0) | 0.048 (10) | No break |
| International soybean price | 0.001 | 0.001 | 0.057 | −15.497 *** (0) | 0.041 (2) | No break |
| U.S. producer price of soybean | 0.001 | 0.006 | 0.069 | −12.185 *** (2) | 0.040 (10) | No break |
| International corn price | 0.001 | 0.001 | 0.058 | −15.910 *** (0) | 0.042 (2) | No break |
| U.S. producer price of corn | 0.001 | 0.001 | 0.075 | −16.719 *** (0) | 0.034 (7) | No break |

Notes: *** denotes rejection of the null hypothesis at the 1% significance level. Numbers in brackets are the lag length and bandwidth. Lag length selection is based on the Bayesian-Schwarz information criterion (henceforth, BIC) in the augmented Dickey-Fuller (ADF) tests. The bandwidth for the KPSS test is determined using the Newey–West bandwidth selection algorithm (Newey and West 1994). We implement all the unit root tests with intercept and trend terms. We used Bai-Perron's sequential test for the hypothesis of *k* breaks versus *k* + 1 breaks, employing the *F*-statistics. Lag length selection was based on the BIC in the test.

## 3. Methodology

We employed a vector error correction[6] generalized autoregressive conditional heteroscedastic model with a multivariate Baba-Engle-Kraft-Kroner specification (VECM-GARCH-BEKK) to scrutinize the associations between the global prices and the U.S. producer prices of major agricultural products. In addition, we used the cross-correlation function as a robustness test that enables us to detect not only directional relationships but also the number of lead-lag relationships as well. First, in order to capture volatility clustering and analyze the volatility spillover effect between the international prices and the U.S. producer prices for wheat, soybean, and corn, the GARCH model with a multivariate BEKK specification was applied. Second, based upon the above results for the unit root tests and cointegration tests, we combined the VECM and asymmetric GARCH-BEKK model. This model has the advantage

---

[2] Dickey and Fuller (1979).
[3] Kwiatkowski et al. (1992) and Phillips and Perron (1988).
[4] The ADF and KPSS unit root tests indicate that all the variables have unit root processes in their levels. We do not report these results for the sake of brevity. The results can be obtained from the authors upon request.
[5] See Bai and Perron (2003).
[6] The characteristic of price series allows for the possibility that there is a station ary long-run equilibrium relationship (cointegrating relationship) between individual price series. If there are no cointegrating relationships among the variables, after *n* (*n* is equal to the order of integrated variables) differences, then the standard vector autoregressive (VAR) model is employed. In contrast, if cointegrating relationships are identified, the vector error-correction model (VECM) can be used for the empirical analysis. Given that all the price series are nonstationary *I* (1) series, we will use the Johansen-Juselius procedure (Johansen and Juselius 1990) to examine the cointegrating relationship between international prices and U.S. producer prices for wheat, soybean, and corn.

of analyzing conditional means and variances to detect dynamic interactions between variables and identifying the direction of causality and spillover effects. More specifically, the VECM can explain price-level behavior by explicitly allowing cointegrated relationships based on the assumptions that the price variances are constant over time. On the other hand, the GARCH-BEKK model considers volatility clustering and captures the patterns of volatility transmissions between pairs of prices. Further, following Grier et al. (2004), our analysis was based upon an asymmetric version of the GARCH-BEKK model, which introduces an asymmetric effect on the model.

Since cointegrating relationships are identified between the pairs of prices, we used the VECM to express the conditional mean equation. Then the representation of a cointegrated system is as follows:

$$p_t = \pi + \gamma EC_{t-1} + \sum_{i=1}^{k} \Phi_i \Delta p_{t-i} + v_t, \; v_t \big| \Omega_{t-1} \sim (0, H_t), \tag{1}$$

where $p_t$ is a $2 \times 1$ vector of monthly returns at time $t$ of the international prices and the U.S. producer prices for each of the three agricultural commodities, and $\pi$ is a $2 \times 1$ constant vector. $EC_{t-1}$ is the lagged error correction term of the cointegration relationship, and $\gamma$ is a $2 \times 1$ vector of parameters that denote the speed of the price adjustment of each equation to long-run equilibrium. Next, $\Phi$ is a $2 \times 2$ matrix of parameters associated with the lagged returns implying individual and cross-causality in means, and $k$ is the order of the lag length. The term $v_t$ is defined as $v_t = (v_{1,t}, v_{2,t})'$ and is conditionally normally distributed with mean vector 0, conditional on past information $\Omega_{t-1}$ and a $2 \times 2$ conditional variance–covariance matrix $H_t$.

Considering the price volatility transmission to positive and negative shocks, we used the asymmetric BEKK model, which has the property that the conditional variance–covariance matrix $H_t$ in Equation (1) is positive definite by structure and can be modeled as:

$$H_t = C'C + A'v_{t-1}v'_{t-1}A + B'H_{t-1}B + D'\omega_{t-1}\omega'_{t-1}D, \tag{2}$$

where $C$ is a $2 \times 2$ lower triangular matrix, and $A$ is the $2 \times 2$ matrix of parameters, which measures the extent to which conditional variances are correlated with past squared errors. $B$ is also a $2 \times 2$ matrix of parameters, this time indicating how the past conditional variance $H_{t-1}$ affects the current levels of the conditional variance. The last item $D'\omega_{t-1}\omega'_{t-1}D$ on the right-hand side introduces the asymmetric effect on the model. Next, $\omega_{t-1} = I[u_{t-1} < 0] \otimes u_{t-1}$, where $I[.]$ is an indicator function equal to 1 if $u_{t-1} < 0$ and 0 otherwise, and "$\otimes$" is the Hadamard product. Finally, $D$ is a $2 \times 2$ parameter matric that measures the potential asymmetric response. Thus, if we expanded Equation (2), the conditional variance equation of the bivariate GARCH (1, 1) model can be represented by

$$\begin{aligned}
\begin{bmatrix} h_{11,t} & h_{12,t} \\ h_{21,t} & h_{22,t} \end{bmatrix} = & \begin{bmatrix} c_{11} & 0 \\ c_{21} & c_{22} \end{bmatrix}' \begin{bmatrix} c_{11} & 0 \\ c_{21} & c_{22} \end{bmatrix} + \begin{bmatrix} a_{11} & a_{12} \\ a_{21} & a_{22} \end{bmatrix}' \begin{bmatrix} v_{1,t-1}^2 & v_{1,t-1}v_{2,t-1} \\ v_{2,t-1}v_{1,t-1} & v_{2,t-1}^2 \end{bmatrix} \begin{bmatrix} a_{11} & a_{12} \\ a_{21} & a_{22} \end{bmatrix} \\
& + \begin{bmatrix} b_{11} & b_{12} \\ b_{21} & b_{22} \end{bmatrix}' \begin{bmatrix} h_{11,t-1} & h_{12,t-1} \\ h_{21,t-1} & h_{22,t-1} \end{bmatrix} \begin{bmatrix} b_{11} & b_{12} \\ b_{21} & b_{22} \end{bmatrix} + \begin{bmatrix} d_{11} & d_{12} \\ d_{21} & d_{22} \end{bmatrix}' \begin{bmatrix} \omega_{1,t-1}^2 & \omega_{1,t-1}\omega_{2,t-1} \\ \omega_{2,t-1}\omega_{1,t-1} & \omega_{2,t-1}^2 \end{bmatrix} \begin{bmatrix} d_{11} & d_{12} \\ d_{21} & d_{22} \end{bmatrix},
\end{aligned} \tag{3}$$

where $h_{ii,t}$ and $h_{jj,t}$ for $i, j = 1, 2$ represent the conditional variances of the returns of each price $i$ and price $j$, respectively, at time $t$. Next, $h_{ij,t}$ is the conditional covariance between price $i$ and price $j$ at time $t$, and $a_{ii}$ and $a_{jj}$ indicate the degree of impact of the current conditional variance (volatility) of price $i$ and price $j$, respectively, from their own past squared errors. Here, $b_{ii}$ and $b_{jj}$ represent the degree of impact of the current conditional variance of price $i$ and price $j$, respectively, from their own past conditional variances. Further, $a_{ij}$ measures the lagged shock transmission from price $i$ to price $j$, and $b_{ij}$ measures the volatility spillover effect from price $i$ to price $j$. Next, $d_{ii}$ and $d_{jj}$ represent the degree of impact of the current conditional variance of price $i$ and price $j$, respectively, from their own past asymmetric shocks (positive or negative), and $d_{ij}$ represents the previous asymmetric shocks transmission from price $i$ to price $j$. If any element in matrix $D$ is positive and significant, then an asymmetric effect exists and negative shocks (bad news) will cause larger volatility in markets than

positive shocks (good news) will. Conversely, a significant negative coefficient in the $D$ matrix implies that bad news may bring less volatility in another market.

Moreover, we used the conditional variances and covariances matrix $H_t$ to generate dynamic correlation coefficients so as to capture the time-varying correlations between global prices and U.S. producer prices in the cereal market. The dynamic correlation coefficient $\sigma_{ij,t}$ between price $i$ and $j$ at time $t$ can be represented as:

$$\sigma_{ij,t} = \frac{h_{ij,t}}{\sqrt{h_{ii,t}}\sqrt{h_{jj,t}}} \; for \; i,j = 1,2. \tag{4}$$

We estimate the parameters of the VECM-GARCH-BEKK model by employing the maximum likelihood estimation method optimized with the Broyden, Fletcher, Goldfarb, and Shanno (BFGS) algorithm. The conditional log-likelihood function $L(\theta)$ can be expressed as:

$$L(\theta) = -T \ln(2\pi) - \frac{1}{2}\sum_{t-1}^{T} \ln|H_t| - \frac{1}{2}\sum_{t-1}^{T} v'_t H_t^{-1} v_t. \tag{5}$$

Furthermore, the structures of the causal relationships between variables are investigated through the Granger causality approach. Specifically, to test for causality in means between international and U.S. producer prices, we used the likelihood ratio statistic to check whether the elements $\phi_{ij}$ of matric $\Phi$ in Equation (1) are significantly different from zero. As for the causality in the variance test, the non-diagonal elements $a_{ij}$, $b_{ij}$, and $d_{ij}$ of matrices $A$, $B$, and $D$, respectively, in Equation (2) were analyzed via the joint Wald test.

## 4. Empirical Results

First of all, the results of the cointegration test strongly suggest the existence of a long-run equilibrium relationship between each pair of agricultural commodity prices,[7] implying that the three prices move together over time. According to these results, this paper employs the VECM instead of a standard VAR model.

Table 2 reports the coefficient estimates for the conditional mean return equation and the conditional variance–covariance matrix of the VECM-GARCH-BEKK model for wheat. First, in terms of the conditional mean estimations, we can observe that both the parameters $\phi_{1,1}$ and $\phi_{2,2}$ are statistically significant, which indicates that the returns of the international and U.S. wheat prices are influenced by their own past returns. In contrast, the coefficients $\phi_{1,2}$ and $\phi_{2,1}$ are statistically significant, which indicates a cross-market (global wheat market and U.S. domestic wheat market) return spillover between lagged and current wheat price returns. In addition, by comparing the absolute values of these coefficients, we can determine that the impacts of the own-market return spillovers are larger than those of the cross-market return spillovers ($|\phi_{1,1}| > |\phi_{1,2}|$ and $|\phi_{2,2}| > |\phi_{2,1}|$). Second, from the estimation results for the variance equation, all the coefficients $a_{1,2}$, $a_{2,1}$, $b_{1,2}$, and $b_{2,1}$ are found to be statistically significant, revealing cross effects. These results suggest that the contemporaneous return and volatility spillovers are bidirectional between international and U.S. producer prices. In other words, lagged shocks and volatility in the international wheat price returns can be exploited to predict future U.S. wheat producer price volatility. Similarly, past changes in the U.S. wheat market could provide valuable information to the global wheat market.

Third, turning to the estimation results for the asymmetric terms $d_{1,2}$ and $d_{2,1}$, we can see that $d_{1,2}$ shows a significant negative value, suggesting that the volatilities of the U.S. producer price are more sensitive to positive shocks (good news) than negative shocks (bad news) in the global wheat market. On the other hand, $d_{2,1}$ has a positive significant value, which provides evidence that the

---

[7]  See Appendix A Tables A1–A3.

arrival of negative shocks in the U.S. wheat market intensifies the global wheat market volatility more than positive shocks of similar magnitude. Finally, the Ljung-Box (Ljung and Box 1978) tests provide evidence that there is no serial correlation in the squared standardized residuals, and the Mcleod-Li test and Lagrange multiplier (LR) test indicate that there is no autoregressive conditional heteroscedastic (ARCH) effect in the wheat model. These diagnostic tests generally support the adequacy of the model specification considered.

**Table 2.** Empirical results of the asymmetric VECM-GARCH-BEKK model for wheat.

| Parameter | International Price ($i = 1$) | | U.S. Producer Price ($i = 2$) | |
|:---:|:---:|:---:|:---:|:---:|
| | Estimate | SE | Estimate | SE |
| $\pi_i$ | 0.124 *** | 0.024 | −0.130 *** | 0.034 |
| $\gamma_i$ | 0.003 *** | 0.001 | −0.003 *** | 0.001 |
| $\phi_{i,1}$ | 0.213 *** | 0.073 | 0.217 *** | 0.062 |
| $\phi_{i,2}$ | 0.129 * | 0.072 | 0.255 *** | 0.062 |
| $c_{i,1}$ | 0.011 *** | 0.003 | 0.017 | 0.003 |
| $c_{i,2}$ | | | 0.005 * | 0.003 |
| $a_{i,1}$ | 0.492 *** | 0.075 | −0.276 *** | 0.096 |
| $a_{i,2}$ | 0.422 *** | 0.077 | −0.074 | 0.129 |
| $b_{i,1}$ | 0.978 *** | 0.031 | −0.192 *** | 0.065 |
| $b_{i,2}$ | 0.086 ** | 0.035 | 0.659 *** | 0.065 |
| $d_{i,1}$ | 0.016 | 0.120 | 0.232 * | 0.122 |
| $d_{i,2}$ | −0.351 *** | 0.120 | 0.721 *** | 0.118 |
| $Q(10)$ | 6.712 | | 15.053 | |
| Mcleod-Li (10) | 11.939 | | 11.370 | |
| LM test | 1.345 | | 1.077 | |

Notes: SE: standard error; *, **, and ***: statistical significance at 10%, 5% and 1% levels.

Moreover, in order to examine the mean and variance causality between the international and U.S. prices for wheat, we perform the joint Wald tests and present the results in Table 3. The results indicate that all the null hypotheses are rejected at the 1% significance level, suggesting a significant bi-directional causality in mean and causality in variance between international wheat price and U.S. wheat producer price. Additionally, the likelihood ratio tests for the null hypothesis that the parameters of the $D$ matrix in the GARCH specification are zero are conducted to test for the relevance of the asymmetric effects. The results of the tests for asymmetric effects show that the null $d_{1,1} = d_{1,2} = d_{2,1} = d_{2,2} = 0$ is clearly rejected at the 1% level, implying strong evidence of asymmetric effects between the global wheat market and the U.S. domestic wheat market.

**Table 3.** Joint Wald tests for causality in mean and variance: wheat.

| Causality in Mean | | | Causality in Variance | | |
|:---:|:---:|:---:|:---:|:---:|:---:|
| Null Hypothesis | Chi-Squared | *p*-Value | Null Hypothesis | Chi-Squared | *p*-Value |
| $\phi_{1,2} = \phi_{2,1} = \gamma_1 = \gamma_2 = 0$ (GP $\leftrightarrow$ LP) | 122.570 (4) | 0.000 | $a_{1,2} = a_{2,1} = b_{1,2} = b_{2,1}$ $= d_{1,2} = d_{1,2} = 0$ (GP $\leftrightarrow$ LP) | 57.277 (6) | 0.000 |
| $\phi_{2,1} = \gamma_2 = 0$ (GP $\nrightarrow$ LP) | 28.141 (2) | 0.000 | $a_{1,2} = b_{1,2} = d_{1,2} = 0$ (GP $\nrightarrow$ LP) | 49.630 (3) | 0.000 |
| $\phi_{1,2} = \gamma_1 = 0$ (LP $\nrightarrow$ GP) | 31.411 (2) | 0.000 | $a_{2,1} = b_{2,1} = d_{2,1} = 0$ (LP $\nrightarrow$ GP) | 24.549 (3) | 0.000 |
| Joint Wald tests for asymmetry effects | | | | | |
| $d_{1,1} = d_{1,2} = d_{2,1} = d_{2,2} = 0$ | 65.677 (4) | 0.000 | | | |

Notes: GP: Global wheat price; LP: Local (U.S.) wheat producer price. The arrow indicates the direction of Granger causality. The likelihood ratio test statistic for causality testing is a $\chi^2(k)$ statistic, where the $k$ within the parentheses is the number of degrees of freedom.

Next, we examine the results for soybean presented in Table 4. First, using a comparison with the findings for the wheat price, the observed mean spillover effect between global and local soybean prices can be reconfirmed. However, the results for soybean prices indicate that the returns for the U.S. soybean price are influenced more significantly past global soybean prices than by its own price returns ($|\phi_{2,1}| > |\phi_{2,2}|$). Second, we also find a uni-directional return spillover between the two prices. Specifically, past shocks in the international soybean price affect the present volatility of the U.S. producer price ($a_{1,2}$ is significant). On the other hand, lagged shocks in the U.S. producer price do not have an impact on the volatility of international soybean prices ($a_{2,1}$ is insignificant). Meanwhile, we can observe that a bi-directional volatility spillover between international soybean price and the U.S. producer price ($b_{1,2}$ and $b_{2,1}$) are both significant. Third, a cross-market asymmetric volatility transmission can be identified. Specifically, the positive significant value of $d_{1,2}$ provides evidence that, compared to positive shocks, negative shocks in the global soybean price cause more volatility in the U.S. producer price. Meanwhile, the negative significant value $d_{2,1}$ suggests that positive shocks to the U.S. producer price have a larger impact on the volatility of the global soybean price. These results are the opposite of the results for wheat. Moreover, the results of the Ljung-Box test, the Mcleod-Li test, and the LM test indicate that the estimated models adequately fit the data.

**Table 4.** Empirical results of the asymmetric VECM-GARCH-BEKK model for soybean.

| Parameter | International Price ($i = 1$) | | U.S. Producer Price ($i = 2$) | |
|---|---|---|---|---|
| | Estimate | SE | Estimate | SE |
| $\pi_i$ | −0.139 *** | 0.048 | −0.474 *** | 0.051 |
| $\gamma_i$ | −0.004 *** | 0.001 | −0.014 *** | 0.001 |
| $\phi_{i,1}$ | 0.778 *** | 0.099 | 1.065 *** | 0.103 |
| $\phi_{i,2}$ | −0.503 *** | 0.071 | −0.760 *** | 0.079 |
| $c_{i,1}$ | 0.033 *** | 0.004 | 0.031 *** | 0.004 |
| $c_{i,2}$ | | | −0.000 | 0.003 |
| $a_{i,1}$ | 0.708 *** | 0.193 | −0.234 | 0.209 |
| $a_{i,2}$ | 0.477 ** | 0.204 | −0.140 | 0.218 |
| $b_{i,1}$ | 0.329 ** | 0.129 | 0.211 ** | 0.094 |
| $b_{i,2}$ | −0.621 *** | 0.121 | 0.959 *** | 0.087 |
| $d_{i,1}$ | 0.717 *** | 0.224 | −0.599 *** | 0.213 |
| $d_{i,2}$ | 0.835 *** | 0.218 | −0.717 *** | 0.186 |
| $Q(10)$ | 9.140 | | 11.548 | |
| Mcleod-Li (10) | 3.491 | | 4.724 | |
| LM test | 0.378 | | 0.426 | |

Notes: SE: standard error; ** and ***: statistical significance at 5% and 1% levels.

Our findings for soybean prices were rechecked with the joint Wald tests reported in Table 5. From Table 5, we can conclude that there exists a bi-directional Granger causality in the means and variances between the international soybean price and U.S. soybean producer price. Further, asymmetry effects are also verified.

Finally, we focus on the empirical results for corn prices presented in Table 6. First, it is interesting to observe that the parameter $\phi_{1,2}$ is statistically insignificant in the conditional mean equation, while its counterpart, $\phi_{2,1}$, is statistically significant. These results indicate a one-way asymmetric return spillover from international corn lagged price to current U.S. corn producer price. Similar to the results for soybean prices, past global soybean price returns have a larger impact on the returns for the U.S. corn price than do its own price returns ($|\phi_{2,1}| > |\phi_{2,2}|$). Second, Table 6 also shows that the coefficients $a_{1,2}$, $a_{2,1}$, $b_{1,2}$, and $b_{2,1}$ are all statistically significant, which provides evidence of a bi-directional return and volatility spillovers between global corn price and U.S. corn producer price. These results are similar to the estimation results for wheat prices. However, the results concerning the asymmetry effects are different from those for wheat and soybean prices. We can see that the asymmetry term $d_{1,2}$ is positive and significant, which suggests that negative shocks from the international corn price cause

more variations than positive shocks in the U.S. corn producer price. In contrast, the estimated value $d_{2,1}$ is not significant, indicating that there is no evidence of asymmetric effect and the shocks from the U.S. corn producer price do not have an impact on the international corn price. Further, all the diagnostic tests show the robustness of our results.

**Table 5.** Joint Wald tests for causality in mean and variance: soybean.

| Causality in Mean | | | Causality in Variance | | |
|---|---|---|---|---|---|
| **Null Hypothesis** | **Chi-Squared** | ***p*-Value** | **Null Hypothesis** | **Chi-Squared** | ***p*-Value** |
| $\phi_{1,2} = \phi_{2,1} = \gamma_1 = \gamma_2 = 0$ <br> (GP $\leftrightarrow$ LP) | 383.601 (4) | 0.000 | $a_{1,2} = a_{2,1} = b_{1,2} = b_{2,1}$ <br> $= d_{1,2} = d_{1,2} = 0$ <br> (GP $\leftrightarrow$ LP) | 45.141 (6) | 0.000 |
| $\phi_{2,1} = \gamma_2 = 0$ <br> (GP $\nrightarrow$ LP) | 323.960 (2) | 0.000 | $a_{1,2} = b_{1,2} = d_{1,2} = 0$ <br> (GP $\nrightarrow$ LP) | 39.103 (3) | 0.000 |
| $\phi_{1,2} = \gamma_1 = 0$ <br> (LP $\nrightarrow$ GP) | 88.770 (2) | 0.000 | $a_{2,1} = b_{2,1} = d_{2,1} = 0$ <br> (LP $\nrightarrow$ GP) | 15.950 (3) | 0.001 |

| Joint Wald tests for asymmetry effects | | | | | |
|---|---|---|---|---|---|
| $d_{1,1} = d_{1,2} = d_{2,1} = d_{2,2} = 0$ | 20.490 (4) | 0.000 | | | |

Notes: GP: Global soybean price; LP: Local (U.S.) wheat producer price. The arrow indicates the direction of Granger causality. The likelihood ratio test statistic for causality testing is a $\chi^2(k)$ statistic, where the $k$ within parentheses is the degrees of freedom.

**Table 6.** Empirical results of the asymmetric VECM-GARCH-BEKK model for corn.

| Parameter | International Price ($i = 1$) | | U.S. Producer Price ($i = 2$) | |
|---|---|---|---|---|
| | **Estimate** | **SE** | **Estimate** | **SE** |
| $\pi_i$ | −0.025 *** | 0.005 | −0.037 *** | 0.006 |
| $\gamma_i$ | −0.015 *** | 0.002 | −0.023 *** | 0.003 |
| $\phi_{i,1}$ | 0.159 * | 0.083 | 0.518 *** | 0.100 |
| $\phi_{i,2}$ | 0.086 | 0.064 | −0.109 | 0.077 |
| $c_{i,1}$ | 0.010 *** | 0.003 | 0.018 *** | 0.005 |
| $c_{i,2}$ | | | 0.008 *** | 0.002 |
| $a_{i,1}$ | −0.147 * | 0.083 | 0.155 ** | 0.064 |
| $a_{i,2}$ | −0.355 *** | 0.126 | 0.475 *** | 0.098 |
| $b_{i,1}$ | 0.928 *** | 0.005 | −0.161 *** | 0.009 |
| $b_{i,2}$ | 0.293 *** | 0.040 | 0.672 *** | 0.063 |
| $d_{i,1}$ | 0.215 ** | 0.107 | −0.076 | 0.084 |
| $d_{i,2}$ | 0.390 ** | 0.186 | 0.008 | 0.170 |
| $Q(10)$ | 3.620 | | 10.207 | |
| Mcleod-Li (10) | 4.040 | | 8.591 | |
| LM test | 0.405 | | 0.881 | |

Notes: SE: standard error; *, **, and ***: statistical significance at 10%, 5% and 1% levels.

In addition, Table 7 reports on the Granger causality tests for corn prices. At the 1% significance level, the results show that bi-directional Granger causality in means and variances is established between the international corn price and the U.S. corn producer price. Similarly, the Wald tests confirm the asymmetric effects between these two price series.

Figure 2 plots the time-varying dynamic correlations between international prices and U.S. producer prices for wheat, soybean, and corn. There are several features worth noting. First, it is interesting to ascertain that all conditional correlations display positive values throughout the entire sample period. This evidence visually confirms that strong co-movements exist between the global prices and the U.S. producer prices in the cereal market. Second, the correlations tend to exhibit extreme variability in some specific time periods. For instance, a substantial decrease in correlation between the global and U.S. soybean markets is apparent in the middle of 1988. Third, the soybean

price has the highest average value (0.878) for its dynamic correlation coefficients compared to the wheat (0.751) and corn (0.836) prices. On the other hand, the wheat price has the highest standard deviation (0.071) compared to the soybean (0.039) and corn (0.047) prices.

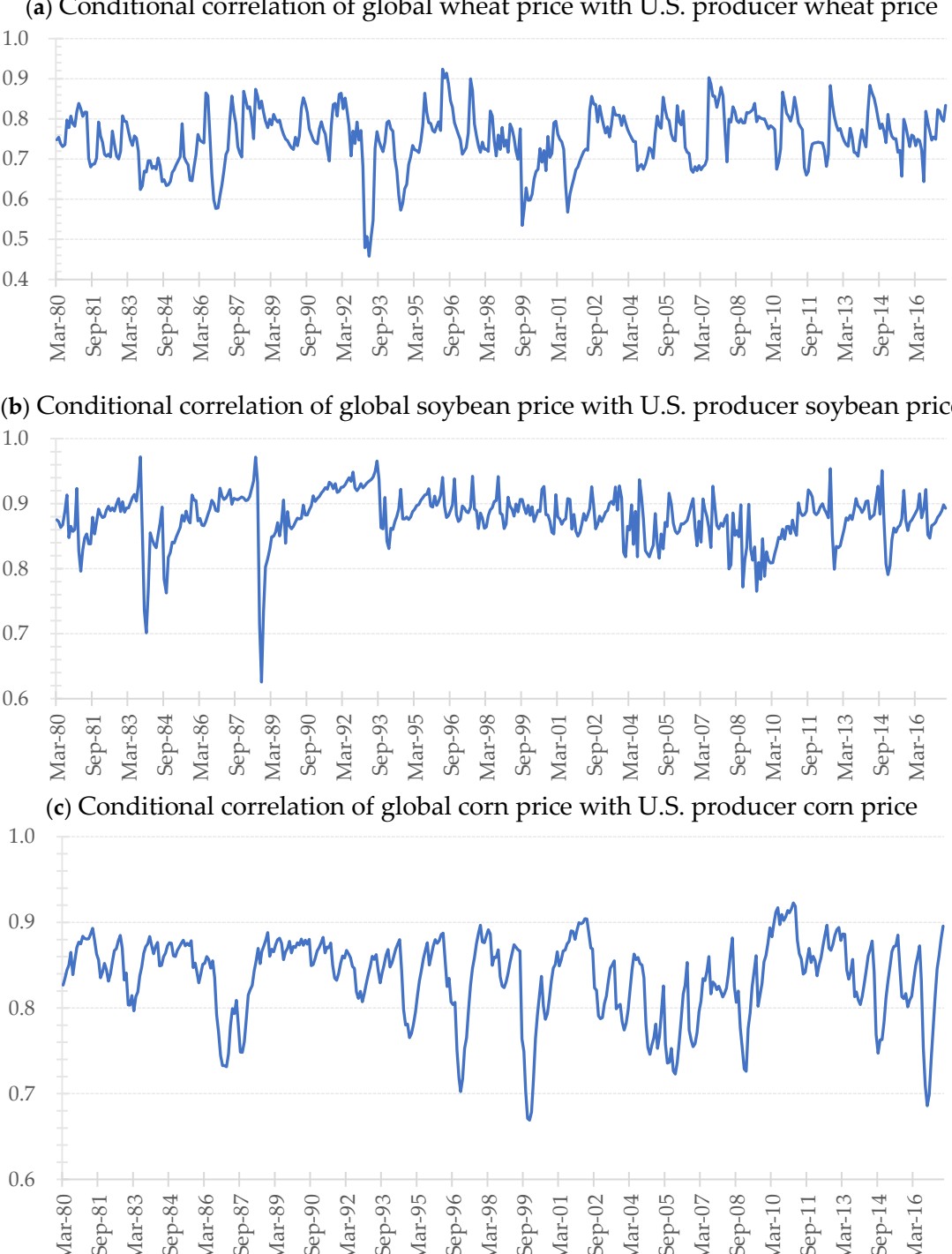

(**a**) Conditional correlation of global wheat price with U.S. producer wheat price

(**b**) Conditional correlation of global soybean price with U.S. producer soybean price

(**c**) Conditional correlation of global corn price with U.S. producer corn price

**Figure 2.** Conditional correlations estimated with the VECM-GARCH-BEKK model.

**Table 7.** Joint Wald tests for causality in mean and variance: corn.

| Causality in Mean | | | Causality in Variance | | |
|---|---|---|---|---|---|
| **Null Hypothesis** | **Chi-Squared** | **$p$-Value** | **Null Hypothesis** | **Chi-Squared** | **$p$-Value** |
| $\phi_{1,2} = \phi_{2,1} = \gamma_1 = \gamma_2 = 0$ <br> (GP ↔ LP) | 190.032 (4) | 0.000 | $a_{1,2} = a_{2,1} = b_{1,2} = b_{2,1}$ <br> $= d_{1,2} = d_{1,2} = 0$ <br> (GP ↔ LP) | 397.089 (6) | 0.000 |
| $\phi_{2,1} = \gamma_2 = 0$ <br> (GP ↛ LP) | 132.134 (2) | 0.000 | $a_{1,2} = b_{1,2} = d_{1,2} = 0$ <br> (GP ↛ LP) | 66.346 (3) | 0.000 |
| $\phi_{1,2} = \gamma_1 = 0$ <br> (LP ↛ GP) | 37.197 (2) | 0.000 | $a_{2,1} = b_{2,1} = d_{2,1} = 0$ <br> (LP ↛ GP) | 369.807 (3) | 0.000 |
| Joint Wald tests for asymmetry effects | | | | | |
| $d_{1,1} = d_{1,2} = d_{2,1} = d_{2,2} = 0$ | 20.250 (4) | 0.000 | | | |

Notes: GP: Global corn price; LP: Local corn price. The arrow indicates the direction of Granger causality. The likelihood ratio test statistic for causality testing is a $\chi^2(k)$ statistic, where the $k$ within parentheses is the degrees of freedom.

## 5. Robustness Check

In the previous section, we found a significant bi-directional causality in mean and causality in variance between international and U.S. producer prices for three crops. This section presents some results on robustness.

Some of the recent studies use multivariate GARCH (MGARCH) models with dynamic conditional correlation (DCC) to analyze volatility spill-over mechanisms.[8] Huang et al. (2010) compare the performances of GARCH-BEKK and GARCH-DCC models and document the superiority of the BEKK model over the DCC model due to the large number of parameters treated by the BEKK model. In contrast, Bauwens et al. (2006) argue that the DCC model outperforms the BEKK model based on a comparison of the goodness of fit. Since the results vary across different datasets and the debate still lacks consensus, we first check the robustness of our empirical results by employing the GARCH-DCC model. The econometric framework of the model was formulated as follows:

$$p_t = \mu + \sum_{i=1}^{k} \varphi_i p_{t-i} + u_t = \mu + \sum_{i=1}^{k} \varphi_i p_{t-i} + \sqrt{H_t} z_t, \tag{6}$$

$$u_t | \Omega_{t-1} \sim N(0, H_t) \tag{7}$$

$$H_t = D_t R_t D_t. \tag{8}$$

Equation (6) is an autoregression (AR) model $k$ process for $p_t$ conditional on the information set $\Omega_{t-1}$. Specifically, $\mu = (\mu_1, \mu_2)'$ is the vector of conditional means, $\varphi_i$ is the parameter vector, $k$ is the lag lengths of the mean equations, $u_t = (u_{1,t}, u_{2,t})'$ is the vector of innovations, $H_t$ is a $2 \times 2$ conditional variance–covariance matrix, $z_t$ is a $2 \times 1$ *i.i.d* vector of standardized residuals, $D_t$ is the diagonal matrix containing the conditional standard deviations on the diagonal, and $R_t$ is the conditional correlation matrix given by:

$$R_t = Q_t^{*-1} Q_t Q_t^{*-1}, \tag{9}$$

---

8    Basher and Sadorsky (2016) and Guo (2018).

where $Q_t$ is the conditional correlation matrix of standardized residuals. $Q_t^{*-1} = diag(q_{11,t}^{-1/2}, q_{22,t}^{-1/2})$ and $q$ is the elements of matrix $Q_t$. Moreover, the matrix $D_t$ can be obtained by estimating a univariate GARCH $(p, q)$ and EGARCH[9] $(p, q)$ model, with $\sqrt{h_{i,t}}$ $(i = 1, 2)$ on the *ith* diagonal as follows:

$$h_{i,t} = \omega_i + \sum_{p=1}^{p_i} \alpha_{i,p} u_{i,t-p}^2 + \sum_{q=1}^{q_i} \beta_{i,q} h_{i,t-q}, \tag{10}$$

$$\ln(h_{i,t}) = \omega_i + \sum_{p=1}^{p_i} (\alpha_{i,p} \left| \frac{u_{i,t-p}}{\sqrt{h_{i,t-p}}} \right| + \kappa_{i,p} \frac{u_{i,t-p}}{\sqrt{h_{i,t-p}}}) + \sum_{q=1}^{q_i} \beta_{i,q} \ln(h_{i,t-q}), \tag{11}$$

where $h_{i,t}$ is a $2 \times 1$ conditional variance vector of the price series and $\omega_i$ is a $2 \times 1$ constant vector, the lag lengths of variance equations are represented as $p$ and $q$. $\alpha$ and $\beta$ are the parameters of the GARCH and ARCH terms, respectively.

Furthermore, Engle (2002) DCC model and Cappiello et al. (2006) asymmetric DCC (henceforth, A-DCC)[10] model were used to determine the volatility spill-over between international and U.S. producer prices for each pair of crops. According to Engle (2002), the dynamic correlation structure is given as:

$$Q_t = (1 - \psi - \zeta)\overline{P} + \psi v_{t-1} v'_{t-1} + \zeta Q_{t-1}, \tag{12}$$

where $Q_t$ is a symmetric positive definite matrix in Equation (8) and $\overline{P}$ is the $2 \times 2$ unconditional correlation matrix of the standardized residuals $v_t$. The parameters $\psi$ and $\zeta$ are non-negative, with a sum of less than unity. Cappiello et al. (2006) modified the correlation evolution equations in the following expression:

$$Q_t = (1 - \psi - \zeta)\overline{P} - \delta\overline{N} + \psi(v_{t-1} v'_{t-1}) + \zeta Q_{t-1} + \delta(\eta_{t-1} \eta'_{t-1}). \tag{13}$$

Equation (13) is a standard A-DCC in which asymmetric terms are included. $\delta$ is the coefficient of the asymmetry term. $\overline{N}$ represents the unconditional matrices of $\eta_t = I[v_t < 0] \otimes v_t$, $I[.]$ is an indicator function equal to 1 if $v_t < 0$ and 0 otherwise, and "$\otimes$" is the Hadamard product. Finally, the parameters[11] of the DCC and A-DCC models were estimated by employing the Gaussian quasi-maximum likelihood estimation (QMLE)[12] with the BFGS[13] optimization algorithm.

First, in order to choose the most suitable GARCH-based models, an extensive specification testing procedure is conducted to ensure that the conditional means and variances processed fit the price series well. In this study, each model is estimated by using the maximum likelihood method, and we determine the lag length of the mean and variance equations based on the BIC. Based on the residual diagnostics and BIC, the AR (1)-EGARCH (1, 1) model was chosen for the international prices for wheat and soybean, and the AR (1)-GARCH (1, 2) model was chosen for the international corn price. On the other hand, for the U.S. producer prices, we chose the AR (1)-EGARCH (1, 2) model, the AR (1)-EGARCH (1, 1) model, and the AR (1)-GARCH (1, 1) model for wheat, soybean, and corn producer prices, respectively.

The estimation results for the GARCH-based models for the international prices and the U.S. producer prices for wheat, soybean, and corn are summarized in Tables 8 and 9, respectively. Each table

---

[9]   The GARCH model was developed by Bollerslev (1986) and the exponential GARCH (EGARCH) model by Nelson (1991).
[10]   The A-DCC model modified the original DCC model by including asymmetries in the correlation dynamics. See Cappiello et al. (2006) for an extensive analysis of these models' advantages.
[11]   The estimated results of the DCC coefficients were not reported for the sake of brevity. The results can be obtained from the authors upon request.
[12]   See Bollerslev and Wooldridge (1992).
[13]   BFGS (Broyden, Fletcher, Goldfarb, and Shanno) is a quasi-Newton optimization method that uses information about the gradient of the function at the current point to calculate where to find a better point.

reports the parameter estimates and their corresponding standard errors. It is noticeable that almost all coefficients of the ARCH and GARCH terms are statistically significant. Meanwhile, the asymmetric terms ($\kappa_i$) capturing the leverage effects are statistically significant for either international price or U.S. producer price for wheat and soybean. These results imply that in these price returns, a negative shock increases volatility more than a positive shock of equal magnitude. Furthermore, the generalized error distribution (GED) is applied in all the models because the GED outperforms the normal distribution. Finally, the results of the diagnostics[14] suggest that the selected GARCH-based model specification explains the data well.

**Table 8.** Empirical results for international price.

|  | Wheat AR (1)-EGARCH (1, 1) | Soybean AR (1)-EGARCH (1, 1) | Corn AR (1)-GARCH (1, 2) |
|---|---|---|---|
| $\theta$ | 0.000 | 0.002 | 0.001 |
| $\varphi_1$ | 0.258 *** | 0.264 *** | 0.233 *** |
| $\omega$ | −0.099 * | −0.245 * | 0.001 *** |
| $\alpha_1$ | 0.054 | 0.036 | 0.013 *** |
| $\kappa_1$ | 0.078 *** | 0.140 *** | - |
| $\beta_1$ | 0.990 *** | 0.964 *** | 1.655 *** |
| $\beta_2$ | - | - | −0.969 *** |
| *GED* parameter | 1.354 *** | 1.195 *** | 1.156 *** |
| $Q(10)$ | 9.087 | 8.270 | 9.707 |
| $Q^2(10)$ | 13.521 | 2.028 | 5.763 |
| ARCH test (10) | 8.367 | 0.225 | 0.552 |
| BIC | −2.964 | −3.085 | −3.038 |

Notes: * and ***: statistical significance at the 10% and 1% levels. The standard error follows Bollerslev and Wooldridge (1992) robust standard error. $Q(10)$ and $Q^2(10)$: test statistics (Ljung and Box 1978) for the null hypotheses of no autocorrelation up to order 10 for standardized residuals and standardized residuals squared. The ARCH test: the Lagrange multiplier test statistic is used to check ARCH effects in residuals (it is distributed as chi-square).

**Table 9.** Empirical results for the U.S. producer price.

|  | Wheat AR (1)-EGARCH (1, 2) | Soybean AR (1)-EGARCH (1, 1) | Corn AR (1)-GARCH (1, 1) |
|---|---|---|---|
| $\theta$ | −0.001 | 0.004 | 0.003 |
| $\varphi_1$ | 0.431 *** | −0.001 | 0.197 *** |
| $\omega$ | −2.119 *** | −0.153 * | 0.003 ** |
| $\alpha_1$ | 0.299 *** | 0.040 | 0.157 * |
| $\kappa_1$ | −0.076 ** | 0.114 *** |  |
| $\beta_1$ | 1.529 *** | 0.977 *** | 0.324 |
| $\beta_2$ | −0.829 *** |  |  |
| *GED* parameter | 1.155 *** | 1.324 *** | 1.189 *** |
| $Q(10)$ | 14.561 | 15.320 | 11.486 |
| $Q^2(10)$ | 13.521 | 8.479 | 14.163 |
| ARCH test (10) | 1.294 | 0.882 | 1.416 |
| BIC | −3.477 | −2.575 | −2.414 |

Notes: *, **, and ***: statistical significance at the 10%, 5%, and 1% levels. The standard error follows Bollerslev and Wooldridge's (1992) robust standard error. $Q(10)$ and $Q^2(10)$: test statistics (Ljung and Box 1978) for the null hypotheses of no autocorrelation up to order 10 for standardized residuals and standardized residuals squared. The ARCH test: the Lagrange multiplier test statistic is used to check ARCH effects in residuals (it is distributed as chi-square).

---

14   The Ljung-Box and ARCH tests show that there is no autocorrelation up to order 10 for the standard residuals and squared standard residuals and no further ARCH effect in all of the models.

Next, we compare the goodness of fit between the GARCH-BEKK and the GARCH-DCC models using the log-likelihood statistics. Table 10 presents the log-likelihoods of three model specifications for wheat, soybean, and corn. We can observe that the BEKK model outperforms the DCC and A-DCC models in terms of higher log-likelihood values. As such, it is reasonable to select the BEKK model as our empirical methodology from the previous section.

**Table 10.** Log-likelihood values for the DCC, A-DCC, and BEKK estimated models.

|  | **Wheat** | **Soybean** | **Corn** |
|---|---|---|---|
| Model | Log-likelihood | Log-likelihood | Log-likelihood |
| DCC | 1572.289 | 1571.507 | 1471.103 |
| A-DCC | 1573.073 | 1573.501 | 1480.206 |
| BEKK | 1651.099 * | 1743.960 * | 1559.138 * |

Notes: * denotes the largest value of log-likelihood.

Finally, we employ the approach proposed by Cheung and Ng (1996) and Hong (2001) to check the robustness of the causality results in mean and variance reported in the previous section. This procedure is based on the residual cross-correlation function (CCF) and is robust to distributional assumptions. The advantage of using the CCF approach is that it detects not only the direction of causality in means and variances but also the number of lead-lag relationships. Specifically, from the estimated GARCH-based model, we calculate the standardized residuals and the standardized squared residuals for international prices and for U.S. producer prices, respectively. Following Cheung and Ng (1996),[15] the sample cross-correlation coefficient at lag $k$, $\hat{r}_{\varepsilon\xi}(k)$ and $\hat{r}_{uv}(k)$ are calculated from the consistent estimates of the conditional means and variances. Under the condition of regularity, we detect the null hypothesis that there is no causality in mean using the following CCF statistic:

$$\sqrt{T}(\hat{r}_{\varepsilon\xi}(k_1),\ \ldots,\ \hat{r}_{\varepsilon\xi}(k_m)) \xrightarrow{L} N(0, \boldsymbol{I_m}), \tag{14}$$

and there is no causality in variance using the test statistic, which is given by:

$$\sqrt{T}(\hat{r}_{uv}(k_1),\ \ldots,\ \hat{r}_{uv}(k_m)) \xrightarrow{L} N(0, \boldsymbol{I_m}), \tag{15}$$

where $k_1,\ \ldots,\ k_m$ are $m$ different integers, and $\xrightarrow{L}$ indicates the convergence in distribution. To test a causal relationship at a specified lag $k$, we compare $\sqrt{T}\hat{r}_{\varepsilon\xi}(k)$ and $\sqrt{T}\hat{r}_{uv}(k)$ with the standard normal distribution. If the test statistic is larger than the critical value for the normal distribution, then we reject the null hypothesis.

The empirical results for wheat are given in Table 11. Lags are measured in months, which range from 1 to 12. From Table 11, we can observe that the test statistic for the null hypothesis that global wheat price does not Granger-cause U.S. producer price in the mean is rejected at lag 1, which indicates that the global price leads the U.S. producer price by approximately one month. Meanwhile, the results display that the causality in mean from the U.S. wheat producer price to the international wheat price exists in lags 3, 10, and 12, respectively. Therefore, there is evidence of a feedback effect in mean between these price pairs. In addition, it is worth noting that mean spillovers from the U.S. producer price to the international price take a longer time than going in the opposite direction in the cross-market information transmission. Turning to causality in variance, the results reveal that the global wheat price Granger-causes the U.S. producer price in variance (in lag 5), with a volatility feedback effect (in lags 2 and 12). These findings imply that the volatility transmission from the U.S. producer price to the global price takes less time than going in the opposite direction.

---

[15]   We did not choose Hong's (2001) method since one of our paper's aims is to detect the number of lead-lag relationships.

**Table 11.** Results for the causality test for wheat.

| Lag $k$ | Mean Causality | | Variance Causality | |
|---|---|---|---|---|
| | International Price → U.S. Price | U.S. Price → International Price | International Price → U.S. Price | U.S. Price → International Price |
| 1 | 2.394 ** | 1.293 | 0.019 | −0.093 |
| 2 | −0.292 | −0.330 | 1.422 | 1.956 * |
| 3 | −0.730 | −1.763 * | 0.345 | 0.093 |
| 4 | 0.256 | 1.048 | −0.959 | 0.226 |
| 5 | 1.183 | 0.353 | 1.649 * | 0.066 |
| 6 | −1.217 | −0.540 | 0.794 | −0.288 |
| 7 | 0.631 | −1.041 | −0.574 | 0.796 |
| 8 | −0.411 | −1.162 | 1.215 | 0.453 |
| 9 | −0.800 | −0.210 | 1.448 | −0.506 |
| 10 | −0.059 | 0.364 | 0.749 | −1.035 |
| 11 | 1.179 | 1.911 * | 0.409 | 1.484 |
| 12 | 0.745 | −1.674 * | 0.478 | 3.141 *** |

Notes: *, **, and ***: statistical significance at the 10%, 5%, and 1% levels. The arrow indicates the direction of Granger causality.

As for the causality test for soybean prices, the results are presented in Table 12. The estimations simultaneously provide strong evidence that there exists either causality in mean or causality in variance between the international soybean price and the U.S. soybean producer price. In particular, the results indicate significant feedback effects in mean and variance that differ in terms of the lag orders. More specifically, we can observe a bi-directional causality in mean with a one-month lag, while the bi-directional causality in variance has about an 11-month lag.

**Table 12.** Results of the causality test for soybean.

| Lag $k$ | Mean Causality | | Variance Causality | |
|---|---|---|---|---|
| | International Price → U.S. Price | U.S. Price → International Price | International Price → U.S. Price | U.S. Price → International Price |
| 1 | 5.776 *** | −2.821 *** | 1.325 | 0.157 |
| 2 | 1.890 * | 0.773 | 0.794 | 0.677 |
| 3 | −1.441 | −1.867 * | −0.150 | −0.148 |
| 4 | −0.400 | 0.210 | −0.764 | −0.809 |
| 5 | −0.938 | −1.812 * | −0.036 | 0.436 |
| 6 | −0.256 | 0.764 | −0.002 | −0.506 |
| 7 | −0.127 | −1.035 | 0.377 | 1.609 |
| 8 | 0.222 | −0.133 | −0.557 | −1.549 |
| 9 | −0.282 | −0.779 | −0.525 | -0.203 |
| 10 | −2.222 ** | −0.616 | 0.356 | 0.984 |
| 11 | 1.118 | 0.402 | 2.620 *** | 1.651 * |
| 12 | 0.019 | −0.034 | 1.564 | 1.854 * |

Notes: *, **, and ***: statistical significance at the 10%, 5%, and 1% levels. The arrow indicates the direction of Granger causality.

Finally, Table 13 reports the results for the causality test for corn prices. According to Table 13, first, we can identify that there is a bi-directional causality in mean and variance between the international corn price and the U.S. corn producer price. Second, similar to the results for wheat, the mean price transmission from the U.S. to international corn price displayed a longer delayed reaction (a five-month lag) than going in the opposite direction (a one-month lag). In addition, we found that the bi-directional volatility transmission between the international and U.S. producer corn prices exhibited the same lag length in both directions (approximately nine months).

**Table 13.** Results of the causality test for corn.

| Lag $k$ | Mean Causality | | Variance Causality | |
|---|---|---|---|---|
| | International Price → U.S. Price | U.S. Price → International Price | International Price → U.S. Price | U.S. Price → International Price |
| 1 | 3.348 *** | 0.237 | −0.694 | −0.182 |
| 2 | 1.441 | −0.510 | −0.377 | 0.392 |
| 3 | 0.436 | 0.275 | 0.275 | −0.787 |
| 4 | −1.090 | −0.032 | 1.456 | 0.135 |
| 5 | −0.607 | −2.982 *** | −0.428 | −0.519 |
| 6 | −1.816 | −0.946 | −0.099 | −0.176 |
| 7 | 0.392 | 1.035 | −1.236 | 0.436 |
| 8 | −2.170 ** | −1.966 ** | −0.351 | −0.707 |
| 9 | 0.152 | 0.324 | 2.038 ** | 2.951 *** |
| 10 | −0.660 | 0.533 | −1.012 | −0.345 |
| 11 | 2.277 ** | 1.577 | 0.881 | 2.570 ** |
| 12 | 0.106 | −1.314 | 0.618 | 0.070 |

Notes: ** and ***: statistical significance at the 5%, and 1% levels. The arrow indicates the direction of Granger causality.

Overall, these empirical results based on the CCF approach are qualitatively similar to those of our VECM-GARCH-BEKK analysis in the previous section. Therefore, our empirical results can be considered robust.

## 6. Conclusions

This paper has explored the Granger causal relationships between international and U.S. producer prices for wheat, soybean, and corn using a VECM-GARCH-BEKK model. Our main findings are as follows. First, a long-run equilibrium relationship exists between the international and U.S. producer prices for wheat, soybean, and corn. Second, significant bi-directional causality in mean and causality in variance are found between the international and U.S. producer prices for the three cereal crops. Third, the results generated in the main analysis using the aforementioned model were endorsed by the CCF sensitivity tests. Finally, the results of the CCF suggest that international wheat and corn prices play a leading role in U.S. local markets in return transmission. Meanwhile, U.S. wheat price can be considered a leading indicator of the global wheat price in volatility transmissions.

It would be meaningful to compare the results from Guo and Tanaka (2019) with ours to contribute to the literature because they also estimated price volatility transmissions between world and local markets in the wheat sector with similar methods. One of the elements to be discussed is causal direction. They found uni-directionality between global and domestic markets, while our results indicated bidirectionality, which is likely to have occurred because Guo and Tanaka (2019) targeted wheat importing regions, whereas we concentrated on a large exporter. This is partly endorsed by An et al. (2016), who investigate price volatility transmissions between domestic wheat and wheat flour prices in Ukraine under export restrictions with a GARCH-BEKK model. The work does not directly analyze the linkage from international to domestic prices, but presents the efficacy of export control policies to insulate domestic from global markets, suggesting that one of the largest exporters in the world may also suffer from tempestuous external markets, which, namely, implies a bi-directional relationship, assuming that Ukraine's export is great enough to influence the world's wheat markets.

Another point of view regards transmission speed. While global wheat price led local prices by one month in our analysis, the counterpart in Guo and Tanaka (2019) estimated it to be five months. This could be caused by the difference between the farm gate price of wheat and the retail price of wheat flour. To deliver wheat flour to consumers, imported wheat needs to be processed at milling factories after trading wheat in wholesale markets, and then to retail stores through wholesale markets of wheat flour. Therefore, the results from Guo and Tanaka (2019) may take five times as long as ours (spillover from international to producer prices).

We found that mean causality from the U.S. producer price to the global price for wheat, corn, and soybean takes more months compared with the opposite direction, which takes just one month. This finding could imply that the U.S. government could have spare time to impose export controls if the country faced a severe poor harvest of those crops to calm domestic markets with increased supply as well as to prevent price transmissions to world markets, which would be beneficial for importing regions of the crops.

It was found that global price also affects prices in significant exporting countries. This finding implies that large exporters need to protect their own local markets from stormy global markets, which is particularly important for low-income households even in advanced economies such as that of the U.S. Our experiments focused on producer price, but consumer or retail price are correlated with producer price movements. As a matter of fact, it is proved that international price volatility of wheat can be transmitted to retail wheat flour prices in exporting countries (Tanaka and Guo 2020). Hence, large crop-exporting governments such as the U.S. government also need to make policy interventions to segregate retail from foreign markets.

We uncovered that U.S. producer price volatility leads to international price volatility in the wheat, soybean, and corn markets, implying that the steadier the U.S. producer prices, the steadier the world prices. The level of productivity (good or bad crops) is the most significant factor leading to changes in producer prices. Therefore, the stability of agricultural production in large exporters is crucial in calming international markets. Hence, investments in weather or climate information sectors with regards to farming could alleviate or lessen price volatility in global markets, consequently improving food security in food-importing countries.

Despite highlighting our analysis on the interlinkages between producer and global prices for three crops, there exists a complex supply chain structure in those sectors. Haile et al. (2017) scrutinized the relationships between the international, wholesale, and retail prices of wheat, flour, and bread in Ethiopia. By encompassing additional explanatory variables such as wholesale price in a model, more detailed connections between markets can be delineated, which is left for future research.

**Author Contributions:** J.G. analyzed and interpreted the data regarding price volatility transmission from international to local markets in U.S. agricultural markets. T.T. performed the background of the paper and discussed the policy implications of the empirical results. All authors read and approved the final manuscript.

**Funding:** The research of the second author is in part supported by a Grant-in-Aid from the Japan Society for the Promotion of Science (Grant Number [A] 18K14533).

**Acknowledgments:** We deeply appreciate anonymous referees for their insightful comments.

**Conflicts of Interest:** The authors declare that they have no competing interests.

**Data Availability:** The datasets generated and analyzed in this study are available in the Global Information and Early Warning System repository: http://www.fao.org/giews/en/. The data that support the findings of this study can be obtained from the corresponding author upon request.

## Appendix A

**Table A1.** Johansen's cointegration test for wheat.

| Null Hypothesis | Maximum Eigenvalue Test | | | Trace Test | | |
|---|---|---|---|---|---|---|
| | Statistic | 5% Critical Value | *p*-Value | Statistic | 5% Critical Value | *p*-Value |
| None | 20.503 | 15.892 | 0.009 | 25.305 | 20.262 | 0.009 |
| At most 1 | 4.802 | 9.165 | 0.306 | 4.802 | 9.165 | 0.306 |

Notes: The lag length of the model is selected as 2 based on the BIC. Test statistics are based on the specifications provided by the constant term. The restriction test on the cointegrating vector is done via the likelihood ratio test. The critical values are computed using Osterwald-Lenum (1992). The *p*-values are based on MacKinnon-Haug-Michelis's *p*-values (Mackinnon et al. 1999).

**Table A2.** Johansen's cointegration test for soybean.

| Null Hypothesis | Maximum Eigenvalue Test | | | Trace Test | | |
|---|---|---|---|---|---|---|
| | Statistic | 5% Critical Value | *p*-Value | Statistic | 5% Critical Value | *p*-Value |
| None | 65.685 | 15.892 | 0.000 | 71.169 | 20.262 | 0.000 |
| At most 1 | 5.485 | 9.165 | 0.234 | 5.485 | 9.165 | 0.234 |

Notes: The lag length of the model is selected as 2 based on the BIC. Test statistics are based on the specifications provided by the constant term. The restriction test on the cointegrating vector is done via the likelihood ratio test. The critical values are computed using Osterwald-Lenum (1992). The *p*-values are based on MacKinnon-Haug-Michelis's *p*-values (Mackinnon et al. 1999).

**Table A3.** Johansen's cointegration test for corn.

| Null Hypothesis | Maximum Eigenvalue Test | | | Trace Test | | |
|---|---|---|---|---|---|---|
| | Statistic | 5% Critical Value | *p*-Value | Statistic | 5% Critical Value | *p*-Value |
| None | 65.685 | 15.892 | 0.000 | 71.169 | 20.262 | 0.000 |
| At most 1 | 5.485 | 9.165 | 0.234 | 5.485 | 9.165 | 0.234 |

Notes: The lag length of the model is selected as 2 based on the BIC. Test statistics are based on the specifications provided by the constant term. The restriction test on the cointegrating vector is done via the likelihood ratio test. The critical values are computed using Osterwald-Lenum (1992). The *p*-values are based on MacKinnon-Haug-Michelis's *p*-values (Mackinnon et al. 1999).

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
