# Peer review of "Dynamic Transmissions and Volatility Spillovers between Global Price and U.S. Producer Price in Agricultural Markets"

_jrfm, doi:10.3390/jrfm13040083_

Round 1

Reviewer 1 Report

Introduction:

The introduction part does not present all relevant references. In particular there are no references about the role of the wholesalers and resellers in the establishment of the international/domestic prices for the cereal markets. Also some space should be devoted to analyse the role and the scope of the world trade regulations versus the domestic agricultural policies in line with the market development.

Data & Methods

The methodology does not capture any influence of the wholesalers’ activity in the volatility and spillover effects between international prices and U.S. producer prices for wheat, soybean, and corn. I think authors should investigate if such influences can be incorporated in the model and also what are the consequences on overall results.

Empirical results:

This section is too long and difficult to be read. The main important findings are hidden behind heavy tables. Although there is a lot of space devote to this part it fails to underlain the possible links between the internal/international market regulations and the main findings.

Author Response

Please find the reply to the review report in the attached file.

Reviewer 2 Report

The paper is well written and the empirical results are interesting. The econometric methodology is not new since it uses well known techniques. The execution is rigorous and the results are well presented. 

Author Response

We thank the referee and editor for their fruitful suggestions. Our responses to the two referees' comments are as follows. A marked-up version of the manuscript with the changes highlighted in the color yellow.

We have checked the robustness of our empirical results by employing GARCH models with a dynamic conditional correlation (DCC). We have also compared the goodness of fit between the GARCH-BEKK and GARCH-DCC models by using the Log-Likelihood statistics. The results of the estimation have been inserted in Section 5.

Reviewer 3 Report

As attached.

Author Response

(The authors gave the same response as above.)

Reviewer 4 Report

This is an interesting and well-performed study. However, my feeling is that in its current form, the manuscript is too focused on the narrow price volatility issue. I understand that this is the primary goal of the study, but I do think the contribution to the broader literature could be improved, there is a potential for that. In view of this, my recommendation is to restructure the paper and refocus the material. 

The Methodology section should aggregate all information related to data and methods. For example, lines 46-54 may be easily integrated with methodology, the repetitions should be avoided. Data-related lines 70-81 may also go the methodology. Starting from line 81 and to the end of section 2 - these are Results, not Methodology (as far as I understood the flow of the text).  

Another recommendation is to discuss the findings, the Discussion section is now missing. A discussion through a prism of other studies in the area will allow the author to emphasize the contribution of the study to the literature.

Author Response

(The authors gave the same response as above.)

Round 2

Reviewer 1 Report

The paper can be accepted for publication.

Reviewer 4 Report

The paper has been improved, the author has addressed most of my recommendations properly